# Pantograph Slider Detection Architecture and Solution Based on Deep Learning

**DOI:** 10.3390/s24165133

**Published:** 2024-08-08

**Authors:** Qichang Guo, Anjie Tang, Jiabin Yuan

**Affiliations:** 1College of Computer Science and Technology, Nanjing University of Aeronautics and Astronautics, Nanjing 210095, China; tychoguo@163.com (Q.G.); jbyuan@nuaa.edu.cn (J.Y.); 2School of Electronic Engineering, Beijing University of Posts and Telecommunications, Beijing 100876, China

**Keywords:** semantic segmentation, deep learning, high-speed railway, pantograph, picture processing, linear array camera

## Abstract

Railway transportation has been integrated into people’s lives. According to the “Notice on the release of the General Technical Specification of High-speed Railway Power Supply Safety Testing (6C System) System” issued by the National Railway Administration of China in 2012, it is required to install pantograph and slide monitoring devices in high-speed railway stations, station throats and the inlet and exit lines of high-speed railway sections, and it is required to detect the damage of the slider with high precision. It can be seen that the good condition of the pantograph slider is very important for the normal operation of the railway system. As a part of providing power for high-speed rail and subway, the pantograph must be paid attention to in railway transportation to ensure its integrity. The wear of the pantograph is mainly due to the contact power supply between the slide block and the long wire during high-speed operation, which inevitably produces scratches, resulting in depressions on the upper surface of the pantograph slide block. During long-term use, because the depression is too deep, there is a risk of fracture. Therefore, it is necessary to monitor the slider regularly and replace the slider with serious wear. At present, most of the traditional methods use automation technology or simple computer vision technology for detection, which is inefficient. Therefore, this paper introduces computer vision and deep learning technology into pantograph slide wear detection. Specifically, this paper mainly studies the wear detection of the pantograph slider based on deep learning and the main purpose is to improve the detection accuracy and improve the effect of segmentation. From a methodological perspective, this paper employs a linear array camera to enhance the quality of the data sets. Additionally, it integrates an attention mechanism to improve segmentation performance. Furthermore, this study introduces a novel image stitching method to address issues related to incomplete images, thereby providing a comprehensive solution.

## 1. Introduction

Railway transportation has been integrated into people’s lives. According to the “Notice on the release of the General Technical Specification of High-speed Railway Power Supply Safety Testing (6C System) System” [1].

Deep learning has been widely used in daily life. In the medical field, Manoj, N.V.S. et al. [2] used deep learning segmentation model to segment medical images to improve the efficiency of distinguishing diseases in the medical field.

Traditional pantograph detection methods mainly use automation technology to obtain the most frequently touched points of the slide block or computer-vision-related methods to draw the wear curve of the pantograph slide block and then use automation-related technology or a manual recognition method to determine whether the slide block needs to be replaced according to these indicators.

Considering the current popular deep learning domain, this paper uses a deep learning semantic segmentation model to segment the edge of the slider. First, the pantograph image is added to the input end by Canny edge detection and then the image on both sides of the pantograph slider is divided by using the improved deep learning model. In this paper, we introduce the attention mechanism as a convolution processing step, which is used to enhance the model’s attention to and representation of the input data. By introducing attention mechanisms, we were able to make the network more focused on the key information in the input data, which improved the perception and performance of the model. Since the bottom of the pantograph is flat and the ground of the image after segmentation is rough, in order to smooth the lower edge of the slide block and obtain the upper edge of the slide block, a new image splicing method is first used for the mask image and then Canny edge detection is carried out for the mask image. The least square method is used for linear fitting of the lower edge of the slide block. The normalized wear curve was drawn. In addition, this paper uses the linear array camera to shoot the data set. The linear array camera has the advantages of high definition, low continuous shooting time interval, not being easily affected by noise, and less interference under strong light and dark lighting conditions. In this paper, the data sets taken by linear array camera and ordinary camera are segmented, respectively, and the segmentation results are compared.

The contributions of this paper are as follows:A pantograph slider data set captured by a linear array camera is constructed, including images processed by Canny edge detection method;A new image stitching method is proposed, which can reduce the influence of noise points on the stitching results and improve the stitching effect.A deep learning network based on an improved semantic segmentation model is designed. Compared with traditional methods, an attention module is introduced as a key innovation. This module can increase the focus on the input data before convolution so that the network can capture important semantic information in the image more efficiently. By introducing the attention mechanism, our model can more accurately distinguish the target object from the background and improve the accuracy and robustness of semantic segmentation. At the same time, the attention module also helps to reduce the sensitivity of the model to irrelevant information, further improving the performance of the network.

## 2. Related Works

The traditional detection methods of pantograph include edge detection, morphological processing and template matching. With the rise in deep learning, the pantograph detection system has also begun to use Convolutional Neural Network (CNN) for pantograph segmentation. S. Gao [3] summarized the current status of pantograph detection and proposed the advancement of deep learning methods, which played an inspirational role in this paper. The traditional detection methods of pantograph include edge detection, morphological processing and template matching. With the rise in deep learning, the pantograph detection system has also begun to use Convolutional Neural Network (CNN) for pantograph segmentation.

In the field of deep learning, detection and segmentation are two important categories. However, although detection can theoretically detect the condition of the pantograph, such as completeness or defect, there is no unified standard and it completely depends on the label specified by the annotator, which is not conducive to the update of the data set or the change in the standard in practical work, and the annotated file may be invalid. We investigated some popular object detection models YOLO [3,4] and found that segmentation is relatively more suitable for pantograph detection tasks.

Minaee, S. et al. [5] comprehensively reviewed the application of deep learning in image segmentation. The authors detail various deep learning methods such as Convolutional Neural Network (CNN), fully convolutional Network (FCN), and Graph Neural Network (GNN) to provide the research ideas for this paper. At the same time, Muduli, D. [6] et al. applied VGG network to the detection of monkeypox skin lesions, which provided confidence for our application to a new neighborhood. Meanwhile, this paper also used VGG16 as the backbone network for comparative experiments. Wei, X. [7] proposed a semantic segmentation model PDDNet, which is of reference significance to this paper.

We investigated some segmentation models: the Unet++ model [8] is a model that uses a nested structure and a redesigned skip connection to achieve more accurate image segmentation and the Unet 3+ model [9] utilizes comprehensive skip connections and deep supervision to improve semantic segmentation performance. However, the Unet model was proposed relatively early, and we investigated Deeplabv3 models, mainly the Deeplabv3 model and Deeplabv3+ model. We found that combining the spatial attention mechanism or multi-scale segmentation spatial pyramid pooling and other methods to add to the segmentation model has a certain effect on the improvement of the segmentation effect [10,11,12,13]. Therefore, we add an attention mechanism module to the Deeplabv3+ model to enhance the accuracy and robustness of the model.

In the Deeplabv3 model, different models can be selected as the backbone network. We investigated the backbone network used by other models, for example, the effect of Resnet variants pretrained on ImageNet (Resnet34, Resnet50 and Resnet101) as feature extractors in the DeepLab model to solve semantic image segmentation tasks [14]. The Deeplabv3+ model was combined with the Resnet50 backbone network for road segmentation [15], and mobilenet was finally selected as the backbone network of this model.

At the same time, we gained inspiration from the work of Li J et al. [16,17] and applied it to the image stitching step [18]. The keypoints are detected and the attention information is added to the keypoints. We also use the idea of this method in the image stitching part, and the attention information is also added to the boundary keypoints of the segmented Mask image.

Jadon, S. A et al. [19] outline various loss functions for semantic segmentation. This paper compares and analyzes the performance, advantages and disadvantages and application scenarios of different loss functions, which provides guidance for the selection of loss functions in this paper.

## 3. Methods

In this paper, labelme image annotation tool is used to annotate the slider contour of the image captured by the linear array camera to obtain the original data set. The traditional pantograph detection method uses Canny edge to detect all the object edges of the image (we will describe the Canny edge detection in detail in a later section) and then extracts the pantograph slider edge separately, which has a reference significance for this paper. Therefore, in the part of the data sets’ enrichment extended data set, this paper adds Canny edge detection images in addition to the images generated by conventional clipping and increasing contrast. Then, the data sets after image processing are input into the deep learning model for training, and other contours of the left and right side of the slide block can be obtained for testing the pantograph image. Since the contour diagram of the slider is not a whole, it is not easy to unify analysis and processing, so the transmission transformation of the contour diagram on both sides is carried out, respectively, to obtain the picture of the front view, and then the splicing is carried out to obtain the complete contour diagram of the slider. (See Figure 1).

### 3.1. Pantograph Image Data Set

Since the shooting environment is on both sides of a high-speed train, this paper uses a line array camera suitable for high-speed movement to collect pantograph pictures. Traditional area array cameras are suitable for static or slow-moving scenes, such as general photography, medical imaging, etc., while the linear array camera adopted in this paper performs well in high-speed motion by continuously capturing images along a line.

The linear array camera, as its name suggests, is in the shape of a “line”. It is also a two-dimensional image, but it is extremely long, a few kilos in length and a few pixels in width. Because of the safety of railway transportation, the pantograph images need to be obtained with high precision; the camera needs to be excited many times by the excitation device, and the multiple “bar” images taken are merged into a huge picture. Therefore, to use a line array camera, it is necessary to use a support line array camera acquisition card.

The working principle of the linear array camera is based on the structure of the sensor arranged in a line. In the process of continuous motion, the optical system focuses the light in the scene onto the line array. The sensor is exposed to the scene light line by line, measuring the light intensity and converting it into an electrical signal. Through timing control, the electrical signals read line by line are transmitted to subsequent data processing stages. Finally, the pixel data converted into a digital signal can be processed to form an image for analysis or display.

The wear degree of a pantograph is divided into two categories: mild wear (1) and severe wear (2). After the linear array camera acquisition, other images that did not contain the pantograph in the original acquisition image were screened and, finally, 208 images were taken from the left and right angles, for a total of 416 pictures; the image size is 2048 × 2048 deep 1 channel depth map. (See Figure 2).

Semantic segmentation can output the categories of each pixel and thus observe the edges of the image. The traditional edge extraction algorithm can only obtain the edge of all objects in the input image but can not obtain the edge of a specific object, but the traditional Canny edge detection algorithm is helpful to the learning of the model, so this article takes Canny edge detection and increases contrast, rotation, shrink and other methods as part of data set enrichment to improve the model learning effect.

The principle of Canny edge detection is mainly to perform differential operation on the image, but the differential process is very sensitive to noise. Therefore, the first step is to use a Gaussian smoothing filter for noise reduction. In short, the current pixel is associated with the surrounding pixel. Using a two-dimensional Gaussian distribution formula, the convolution kernel with Gaussian distribution weight is obtained and then the weight of the convolution kernel is normalized to achieve image denoising.
(1)$$G(x,y)=\frac{1}{2\pi \sigma^{2}}e^{-(x^{2}+y^{2})/2\sigma^{2}}$$

In the second step, the Sobel operator is used to calculate the gradients and angles of the image along the x and y directions. Any pixel only has adjacent pixels in the upper, lower, left, right and positive oblique directions, so the image is suppressed according to the range of non-maximum values. If or is compared with the left and right pixels and the pixel is the maximum value, it is retained; otherwise, it is set to 0. Similar applies if or is compared to the upper and lower two pixels; if or is compared with the top left and bottom right two pixels; and if or is compared with the bottom left and top right two pixels.
(2)$$S_{x}=\left[\begin{array}{ccc} -1 & 0 & 1\\ -2 & 0 & 2\\ -1 & 0 & 1 \end{array}\right],S_{y}=\left[\begin{array}{ccc} 1 & 2 & 1\\ 0 & 0 & 0\\ -1 & -2 & -1 \end{array}\right]$$

Third, high and low thresholds are used to reduce pseudo-edges. If the amplitude of a pixel position exceeds the high threshold, the pixel is retained as an edge pixel. If the amplitude of a pixel position is less than the low threshold, the pixel is excluded. If the amplitude of a pixel position is between two thresholds, it is classified as edge or non-edge according to connectivity: if the pixel is adjacent to the pixel identified as edge, it is judged as edge; otherwise, it is non-edge.

After adding Canny edge detection images to the original data set, the final data set is obtained.

### 3.2. Split the Pantograph Slider

Attention mechanisms have become very popular recently to improve the focus of models on specific features. We surveyed many papers on attention mechanisms. Hou, Q. et al. [20] proposed a new Attention mechanism called “Coordinate Attention” to improve the efficiency of mobile networks. The proposed mechanism achieves accurate feature capture and transfer by introducing coordinate information into channel attention.

In this paper, the SE attention module is added before convolution to improve the expressiveness of the model [21]. The attention module is divided into two parts, channel attention and spatial attention, which calculate the weights corresponding to channel importance and spatial importance, respectively.

The attention mechanism is introduced into the semantic segmentation of pantograph in railway transportation, and the recognition ability of pantograph details is improved by dynamically adjusting the model’s attention to different features. This not only enhances the segmentation accuracy of the model in complex backgrounds but also improves the overall robustness and generalization ability, which is helpful to detect and locate the pantograph more accurately in practical applications.

The channel attention module first maximizes and average pools the feature graphs of size (C,W,H), respectively, to make their sizes become (C,1,1); then, convolution is performed to reduce the number of channels by 16 times, the ReLU function is used to increase the nonlinearity and, finally, convolution is performed again to recover the number of channels. Through Sigmod function, the final output out=avg_out+max_out represents the importance of the channel weight.

The spatial attention module first obtains the maximum value and average value of all dimensions at a certain point in the corresponding plane of the feature graph and then concatenates in the direction of the channel to obtain the feature graph of (2*W*H). Then, after convolution, keeping W and H unchanged, the channel dimension becomes 1 and the importance weight of the space is obtained through a Sigmod function. In the last step, the channel attention weight and space attention weight are multiplied, respectively, on the original feature map to obtain the result after adding CBAM module.

The following Figure 3 is the model structure diagram of this paper.

### 3.3. Segmentation Image Processing

In this section, the specific methods of processing the two segmented images are deeply discussed, and the methods of image affine transformation, stitching and analysis are mainly introduced to improve the accuracy and effectiveness of the wear detection of the pantograph.

#### 3.3.1. Transmission Transformation

Pantograph shooting usually requires a front image to be obtained in order to facilitate image processing. However, due to practical conditions such as the difficulty of mounting the camera on the front, the side-shot solution still exists. Images taken from the side cannot be processed directly and need to be transferred to the front using transmission transformation. The method used in this paper is the side shot, that is, the calibration plate is set up during the shooting and, in the subsequent processing, the co-ordinates of the four points on the calibration plate are marked and the image is transformed into a rectangle through transmission transformation.

The perspective transformation study is the relationship between co-ordinate changes. This type of transformation does not preserve the parallelism, length and angle of the information; the perspective transformation formula is as follows.
(3)$$\left[x^{\prime },y^{\prime },z^{\prime }\right]=\left[u,v,w\right]\left[\begin{array}{ccc} a_{00} & a_{01} & a_{02}\\ a_{10} & a_{11} & a_{12}\\ a_{20} & a_{21} & a_{22} \end{array}\right]=\left[\begin{array}{cc} T_{1} & T_{2}\\ T_{3} & a_{22} \end{array}\right]$$
where (u, v) is the co-ordinate of the original image, w takes the value 1, and (x = x′/z′, y = y′/z′) is the result of the transmission transformation. The latter matrix is called the transmission transformation matrix, where $T_{1}$ represents the linear transformation of the image, $T_{2}$ represents the translation of the image, $T_{3}$ represents the projection transformation of the image and $a_{22}$ is usually set to 1. The $a_{ii}$ is the transformation matrix.

#### 3.3.2. Image Stitching Method with Bottom Edge Alignment

The front view angle of the left and right pantograph images can only extract fragments of the pantograph wear information, so the bottom edge of the left and right front view angle is aligned first and then the image is spliced.

First, it is detected vertically from the lower right corner of the left image and the lower left corner of the right image respectively. If the gray level of the pixel co-ordinate (x, y) (the gray level ranges from 0 to 255) and the gray level of 10 pixels up the y axis are different by 50, then the co-ordinate is defined as the demarcation point. Considering that there may be noise in the image, this paper reduces the number of 10 pixels detected upward to 7, that is, as long as 7 pixels in the upper 10 pixels meet the gray difference of 50, it is determined to be the lower edge of the pantograph slider.

After the bottom edge is determined, a new image is created with the color black, the height is the maximum of the height of the two diagrams and the width is the sum of the width of the two diagrams. The left picture is pasted to the top left of the picture, that is, the starting position is (0, 0), the x co-ordinate of the right picture translates the width of the left picture and the y co-ordinate is set to make the lower edges of the two pictures coincide, that is, the starting co-ordinate is:(4)$$\left(x_{left}+1,y_{left}-y_{right}\right)$$

Compared with the traditional method, which only determines the gray level of position pixels, the splicing method adopted in this paper has a better effect.

#### 3.3.3. Linear Fitting of the Bottom of the Slider by Least Square Method

The division result of the lower edge of the pantograph slider is uneven and the bottom of the pantograph slider does not touch the wire in practice, so it is basically complete. Therefore, in this paper, the least square method is used to synthesize the lower edge of the pantograph slider into a straight line.

In this paper, it is assumed that the fitted line is y = B. For the co-ordinates of the lower edge of the pantograph image slider, the least square formula is used to obtain *b* to make it the minimum, that is:(5)$$L={\sum_{i=1}^{n}\left(y_{i}-b\right)}^{2}$$

Let the partial derivative of the above equation be 0, so as to obtain the optimal solution of the parameters.
(6)$$\frac{\partial L(b)}{\partial b}=-2\sum_{i=1}^{n}\left(y_{i}-b\right)=0$$

#### 3.3.4. Calculation of Normalized Edge Curves

In this paper, by observing the comparison between the normalized minimum wear and the maximum wear, we can determine whether the slider needs to be replaced. First, calculate the distance between the bottom and top of each corresponding pantograph slider.

Secondly, calculate the distance d=|y−b| of all edge points at the bottom and top of the pantograph slide, where y is the ordinate of the upper edge and b is the obtained co-ordinate of the simulated straight line at the lower edge of the pantograph slide. In other words, d is the distance between the bottom and top of the pantograph slide corresponding to the edge points in the vertical direction.

Thirdly, in order to intuitively represent the wear degree of the skateboard, the thickness of the skateboard is normalized according to the integrity of the skateboard (that is, the minimum wear):(7)$$d_{n}=\frac{d}{d_{\max}}$$
where $d_{\max}$ is the maximum distance from the bottom to the top (the original thickness of the pantograph slide). (See Figure 4).

Finally, the wear curve of the skateboard is plotted based on all calculated $d_{n}$.

## 4. Experiments and Simulation Results

In this paper, the image segmentation method of the slide block of a train pantograph is studied. Before input, we deal with the images; the image after Canny edge detection is added and then labeled. The image is then segmented using a deep learning model. Then, the transmission transformation of the pantograph segmentation results is carried out, and the relative height of the slide block is calculated. Therefore, we obtain the wear state diagram of the pantograph slide block. Finally, the data sets of the linear array camera used in this paper and the ordinary camera are trained and the segmentation results are compared.

In order to test the segmentation effect of the method and the advantages of the linear array camera used compared with the conventional camera, two indicators were quantitatively evaluated: the mean crossover ratio (MIoU) and the class-average pixel accuracy. They are explained in the following two formulas.

MIoU

Intersection ratio (IoU) is one of the commonly used evaluation indexes in semantic segmentation tasks. It represents the ratio of the intersection between the real part of an image and the predicted part of an image:(8)$$IoU=\frac{TP}{TP+FN+FP}$$
where TP is the intersection of the real part and the prediction part, and TP + FN + FP is the union of the real part and the prediction label.

MIoU is the average of the intersection ratio of each class in the data set, calculated as follows:(9)$$MIoU=\frac{1}{k+1}\sum_{i=0}^{k}\frac{p_{ii}}{\sum_{j=0}^{k}p_{ij}+\sum_{j=0}^{k}p_{ji}-p_{ii}}$$
where $p_{ij}$ indicates that category i is predicted to category j, and k indicates the number of categories to be divided; usually the background is treated as a category so the denominator is k + 1, and the above equation is equivalent to the following.

The closer the value of IoU is to 1, the better the training effect is.

Category Average Pixel Accuracy (MPA)

MPA is the average of the sum of all categories of pixel accuracy. First, the pixel accuracy of each class is obtained and then they are summed and then averaged.
(10)$$MPA=\frac{1}{k+1}\sum_{i=0}^{k}\frac{TP_{i}}{TP_{i}+FP_{i}}$$

This paper selects three models, Unet, Deeplabv3 and PSPnet, and uses different backbone networks to carry out comparative experiments with the network in this paper.

The comparison results show that, compared with other semantic segmentation models, the proposed algorithm has higher segmentation accuracy. The MIoU of this algorithm reaches 92.32, which is 18.79%, 5.25%, 2.04% and 0.7% higher than PSPnet, Deeplabv3plus/xception, Unet and Deeplabv3plus/mobilenet, respectively. MPA increased by 4.356% and 0.093%, respectively. The IoU of the pantograph slider was 85.46, which increased by 36.29%, 4.18%, 10.33% and 2.3%, respectively. MPA is higher than Deeplabv3plus/xception and PSPnet. (See Table 1).

The results of the five models after 100 rounds of training are as follows. (See Figure 5).

As can be seen from the figure, the segmentation effect of Deeplabv3plus/xception and PSPnet is not very good, mixed with some background and many areas are not completely segmented. Unet and Deeplabv3plus/xception models are effective in detecting the upper and lower edge of the slider, but the fluctuation of the left and right starting points is large, which is not conducive to the subsequent depth calculation. In general, the edge detection results of this model are the best.

Linear array cameras still have good performance in extreme environments such as strong or low light, rainy weather and other conditions. Compared with ordinary cameras, linear array cameras have the characteristics of short shooting delay, high resolution and low noise image noise. In this paper, by simulating the pictures taken by ordinary cameras, the segmentation effect is compared with that of linear array cameras.

In this paper, the data set taken by the linear array camera is blurred, noisy and reduced, and a total of 100 images of ordinary images and 266 images of linear array camera images are obtained. (See Figure 6).

Deeplabv3+ model of the mobilenet backbone network was used to train ordinary images and line array camera image segmentation data sets, respectively, and their MIoU and MPA were compared. The results are shown in the following Table 2.

As shown in the table, the linear array camera achieves a 2.1% MIoU improvement over the normal camera. This enhancement is particularly significant in the pantograph slider category, where the MIoU increases by 3.94%. In addition, the MPA is also increased by 4.2%.

This improvement can be attributed to several features of linear array camera technology. Firstly, the linear array camera provides higher spatial resolution and higher image clarity, which plays an important role in accurately capturing the details of the pantograph slider, and this increase in resolution helps to enhance the MIoU by providing a more accurate division of object boundaries. (See Figure 7).

The above are the results of segmentation after 100 rounds of Unet training for three groups of linear array cameras and ordinary cameras. It can be seen that the effect of ordinary cameras is obviously not as good as that of data sets captured by linear array cameras due to the influence of resolution, lighting environment and other factors.

## 5. Conclusions

The quality of the pantograph slider has an important impact on the normal operation of the train and the stable supply of power. The traditional replacement method requires a lot of manpower and material resources and it is easy to receive the promotion of complex scenes such as weather, illumination and seasonal changes, so it is difficult to meet the actual needs. In this paper, a pantograph data set of a high-speed railway train captured by linear array camera and ordinary camera is established to improve the segmentation effect of the image as much as possible. The pantograph sliding block segmentation method designed in this paper also achieves good results on the data set. Experiments show that the network has a great improvement over the traditional methods. By adding an additional Canny edge detection data set and adding an attention module, it can be applied to the actual scene of train slide state detection, improve the segmentation effect of a high-speed train pantograph slide and have broad application prospects.

## Figures and Tables

**Figure 1 sensors-24-05133-f001:**
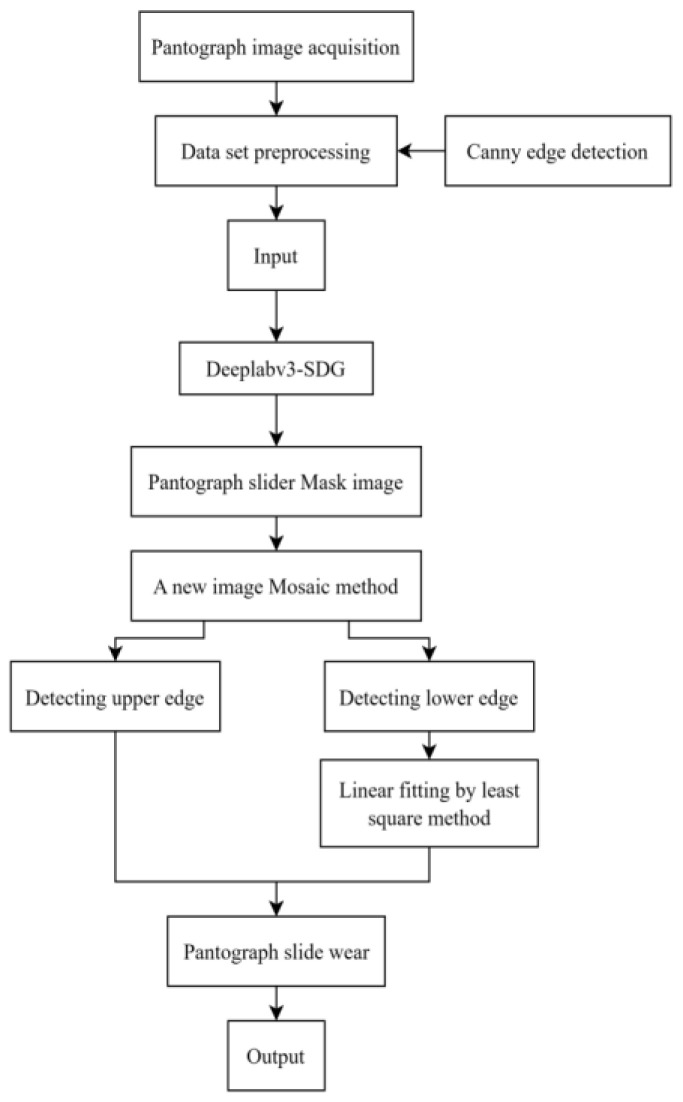
Pantograph image processing flow chart.

**Figure 2 sensors-24-05133-f002:**
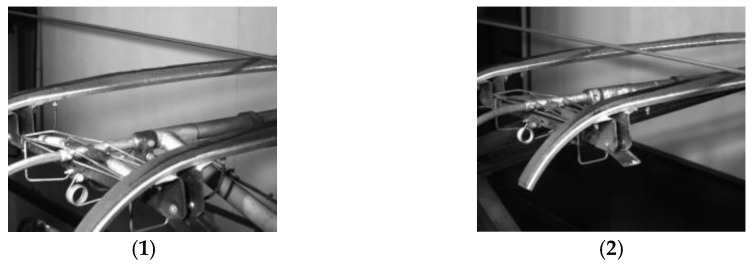
Schematic diagram of data set. The left and right images are the parts that need to be segmented before and after the pantograph slider, which are taken in practice using a linear array camera.

**Figure 3 sensors-24-05133-f003:**
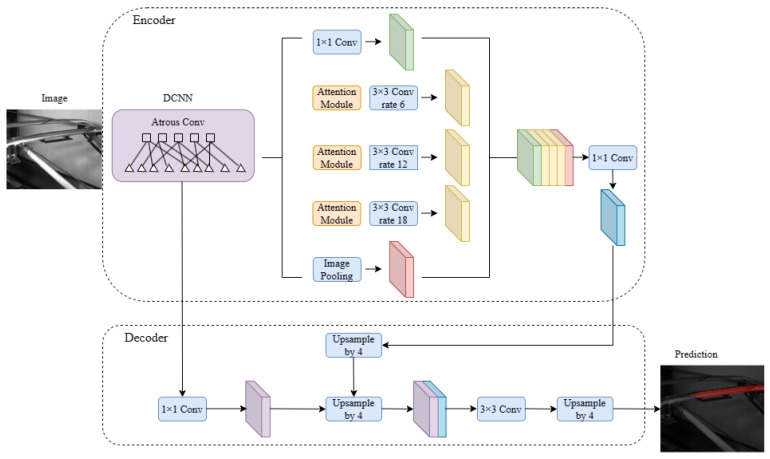
Schematic diagram of the model in this paper.

**Figure 4 sensors-24-05133-f004:**
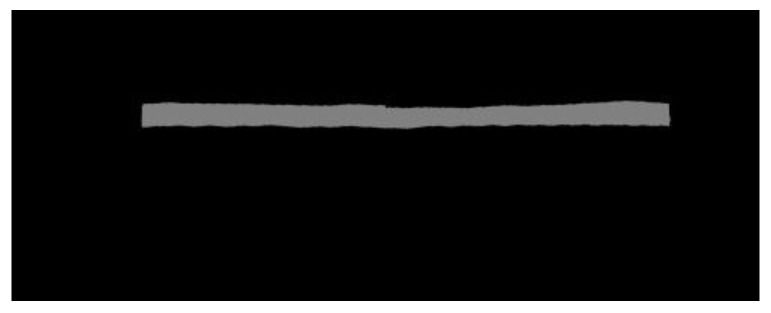
Stitching result.

**Figure 5 sensors-24-05133-f005:**
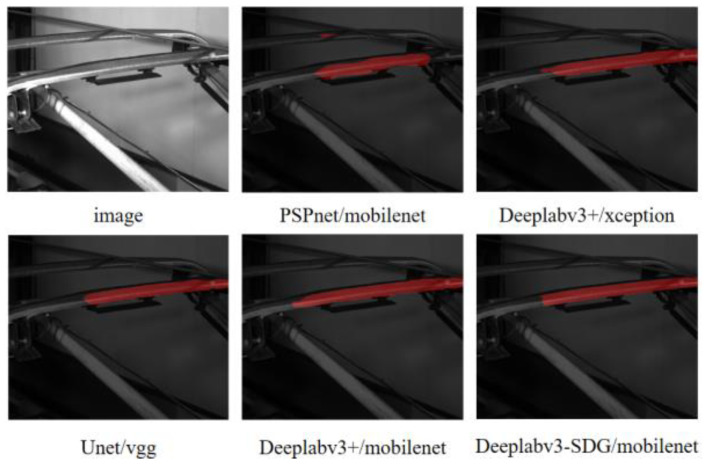
Segmentation results for different models and backbones.

**Figure 6 sensors-24-05133-f006:**
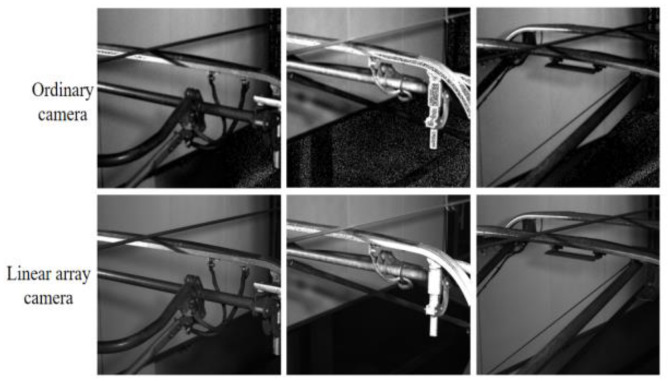
Comparison of photos taken by a linear array camera and an ordinary camera.

**Figure 7 sensors-24-05133-f007:**
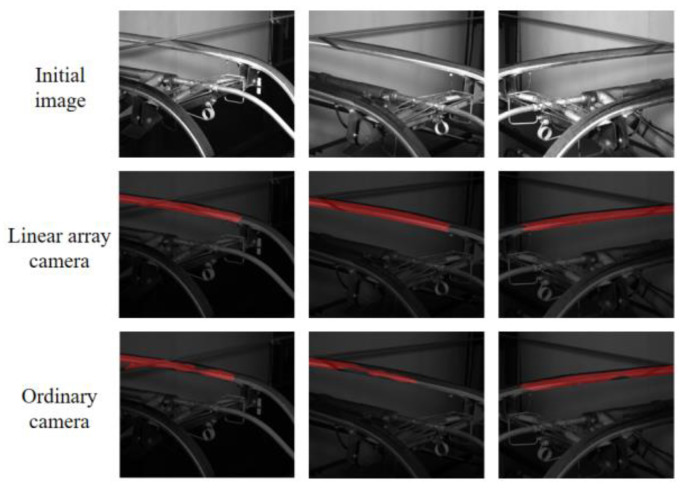
Comparison of segmentation results between linear and area array cameras.

**Table 1 sensors-24-05133-t001:** Experimental results of each model slider segmentation.

Model	Backbone	Category		MIoU	MPA
		Background	SDG		
**PSPnet**	**mobilenet**	97.89	49.17	73.53	79.09
**Deeplabv3plus**	**xception**	99.27	81.28	87.07	92.16
**Unet**	**vgg**	99.02	75.13	90.28	94.96
**Deeplabv3plus**	**mobilenet**	93.26	83.16	91.62	94.33
**Deeplabv3-SDG** **(This model)**	**mobilenet**	99.17	85.46	92.32	93.87

**Table 2 sensors-24-05133-t002:** Results of linear array camera and ordinary camera segmentation.

Camera Category	Category		MIoU	MPA
	Background	SDG		
**Ordinary camera**	**99.02**	**77.34**	**88.18**	**90.76**
**Linear array camera**	**99.27**	**81.28**	**90.28**	**94.96**

## Data Availability

The data are unavailable due to privacy or ethical restrictions.

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
