# Peer review of "Pantograph Slider Detection Architecture and Solution Based on Deep Learning"

_sensors, 2024, doi:10.3390/s24165133_

Round 1

Reviewer 1 Report

Comments and Suggestions for Authors

This paper studies pantograph slider segmentation, and a deeplabv3 model is used for this purpose. Overall, this work lacks technical novelty, and the manuscript is poorly organized and written. My comments are as follows.

1) The authors do not cite any references in the introduction, which is inappropriate. Also, the background and significance of pantograph slider detection should be highlighted.

2) The authors summarize a lot of works on Sec. Related works. However, there seems to be no regularity and it is merely a stack of existing works. It does not point out the problems in the current work.

3) In the experimental part, a comparison with the latest semantic segmentation models is required,  the author only compared with PSPNet, U-Net, and DeepLabv3+, which is insufficient.

4) It is better to revise Figure 3. As far as I know, this figure is first presented in DeeplabV3+.

5) This paper is not well written, and I could only consider this work as an experimental report, not a research paper.

Comments on the Quality of English Language

It must be improved. There are many grammar errors and typos.

Reviewer 2 Report

Comments and Suggestions for Authors

This paper introduces a pantograph state recognition method based on semantic segmentation, and introduces an attention mechanism to increase the model effect. At the same time, the pantograph splicing method in this paper has certain innovation. I suggest some revision. The specific review comments are outlined below:

1.       As the focus of this paper, the analysis of the attention mechanism method is not enough space, and it is suggested that the author make appropriate additions to the analysis part.

2.       When Canny edge detection is mentioned, the method is not introduced and analyzed, but introduced in the subsequent chapters brusque. It is suggested to introduce specific chapters in the beginning.

3.       The authors should furnish detailed introductions for the datasets or specify their source in the paper.

4.       There is less content in the experimental analysis section, especially in the comparison section between linear and area array cameras, which has only a short one-line description without in-depth analysis, which should be supplemented as a core reader of the paper.

5.       In Experiment 1, although the performance of MIoU is improved, the MPA is decreased. Please explain the reason for the decrease of MPA.

In conclusion, while the paper presents valuable contributions, it requires some revisions to address the identified issues and enhance the overall quality.

Comments on the Quality of English Language

English language needs check some typos. 

Reviewer 3 Report

Comments and Suggestions for Authors

Strong aspects:

1. The Canny edge detection datasets is added as the input to train the model, which has a certain heuristic effect on improving the semantic segmentation effect.

2. The spatial attention mechanism and channel attention mechanism are added to improve the accuracy and robustness of semantic segmentation.

3. Based on the semantic segmentation model, spatial attention mechanism and channel attention mechanism are added to improve the accuracy and robustness of semantic segmentation.

Weak aspects:

1.In Section 3.1, this paper introduces the principle of linear array camera and application scenarios, but does not explain the scene of the paper itself, and how the linear array camera fits into the scene, which will cause confusion to the readers. We encourage the author to increase the description of the camera in the scene of this paper, and the way the linear array camera fits the scene.

2.The attention mechanism added as the core of this paper is not enough space, so the author is recommended to improve or introduce it in detail.

3.Only the algorithm is presented in the stitching section, and we encourage the authors to present concrete stitching results.

4.It is recommended to provide more detailed notes and explanations when presenting graphs and data so that readers can better understand the experimental design and results.

5.For some key results, more background and contextual explanations are provided to help the reader understand the reasons behind the results.

In my opinion this article can be accepted after minor revisions.

Comments on the Quality of English Language

The quality of English is good.

Reviewer 4 Report

Comments and Suggestions for Authors

The author proposed a pantograph detection model, which has certain innovation and increases the attention mechanism to improve the accuracy and robustness of detection. However, there are still some problems in this paper that need to be addressed. The specific review comments are as follows:

1. Although the effect of the linear array camera is better than that of the ordinary camera, the authors did not analyze the results, and it is recommended that the authors add some space in the second experimental section.

2. There is a misspelling on line 363. "segmnetation" should be changed to "segmentation".

3. The algorithm introduction of the splicing part is too simple and not detailed enough. The length should be increased.

4. The formula format in the paragraphs of the article is different, and the author is advised to modify it, such as line 197 "(2*W*H)", line 190 "(C, W, H)", "(C, 1,1)".

5. The variable aii in the equation (3) is not specified in detail and should be added by the author.

In my opinion, the article can be accepted after minor revisions.

Comments on the Quality of English Language

I suggest the author further polish the article.

Round 2

Reviewer 1 Report

Comments and Suggestions for Authors

1)In the abstract, too much text is focused on the background, and it should emphasize the current issues of the existing methods and the proposed solutions.

2)You should draw a conclusion at the end of the abstract.

3)You did not cite any references in the Sec—introduction, which is inappropriate.

4) The sentence 'S.Gao[8] summarized the current status of pantograph detec-75 tion and proposed the advancement of deep learning methods, which played an inspira-76 tional role in this paper.' has appeared twice.

5) In the Sec-related work, the author did not introduce research work related to pantograph slider detection, such as traditional methods and CNN-based methods.  This section is very confusing.

6) Please number the references starting from 1.

7)It seems that the authors utilized SE attention (channel attention) in the model. However, the authors did not cite the paper SENet.

8) More deep segmentation methods should be compared to highlight the superior performance of the proposed model.

Comments on the Quality of English Language

Line 54, '1.A' -> '1. A'

Line 56, '2.A' -> '2. A'

Round 3

Reviewer 1 Report

Comments and Suggestions for Authors

Most of my concerns have been addressed.

Comments on the Quality of English Language

NA.